# Plasma Clearance of Intravenously Infused Adrenomedullin in Rats with Acute Renal Failure

**DOI:** 10.3390/biom12091281

**Published:** 2022-09-11

**Authors:** Hiroshi Hosoda, Tsutomu Nakamura, Fumiki Yoshihara

**Affiliations:** 1Department of Hypertension and Nephrology, National Cerebral and Cardiovascular Center Hospital, 6-1 Kishibe-Shimmachi, Suita 564-8565, Osaka, Japan; 2Department of Molecular Pathophysiology, School of Medicine, Shinshu University, 3-1-1 Asahi, Matsumoto 390-8621, Nagano, Japan; 3Education and Research Center for Clinical Pharmacy, Faculty of Pharmacy, Osaka Medical and Pharmaceutical University, 4-20-1 Nasahara, Takatsuki 569-1094, Osaka, Japan

**Keywords:** adrenomedullin, plasma clearance, calcitonin receptor-like receptor, receptor activity modifying proteins, kidney

## Abstract

Plasma adrenomedullin concentrations are reportedly elevated in patients with renal failure; however, the underlying mechanism is unclear. In this study, we investigated the plasma clearance of synthetic human adrenomedullin (AM) in two models of rats with renal dysfunction; one was induced by subcutaneous injection of mercury chloride (RD-Ag) and the other by completely blocking bilateral renal blood flow (RD-Bl). Sixty minutes after starting intravenous AM infusion, AM levels in RD-Ag, RD-Bl, and rats with normal renal function (NF) were still increased slightly; however, plasma AM levels in RD-Ag rats were approximately three times as high as in RD-Bl and NF rats. Plasma AM disappearance after the end of treatment was similar among the three groups. Pharmacokinetic analysis revealed that elevated plasma AM in RD-Ag rats may be caused by a reduced volume of distribution. The adrenomedullin functional receptor is composed of heterodimers, including GPCR, CLR (calcitonin receptor-like receptor, CALCRL), and the single transmembrane proteins, RAMP2 or RAMP3 (receptor activity modifying protein). *Calcrl* expression was downregulated in the lungs and kidneys of RD-Ag rats. Furthermore, the plasma concentration of exogenous AM was elevated in mice deficient in vascular endothelium-specific *Ramp2*. These results suggest that decreased plasma AM clearance in RD-Ag is not due to impaired renal excretion but to a decreased volume of distribution caused by a reduction in adrenomedullin receptors.

## 1. Introduction

Adrenomedullin (AM) is an antihypertensive peptide isolated from human pheochromocytoma tissue by monitoring the elevating activity of platelet cAMP [1]. The 52-residue peptide has one intramolecular disulfide bond and is subject to C-terminal amidation. AM shows a small degree of sequence homology with calcitonin gene-related peptide (CGRP) and exerts a potent and long-lasting hypotensive effect similar to that of CGRP [1,2,3]. Human adrenomedullin mRNA is highly expressed in several tissues, including adrenal medulla, ventricle, lung, kidney, and vessel walls as well as pheochromocytoma, adrenomedullin peptide circulates in the blood at a considerable concentration [4,5]. Mice deficient in the *Adm* gene are embryonic lethal due to disruption of blood vessels and consequent intra-organ and subcutaneous hemorrhage followed by circulatory collapse [6,7]. AM is indispensable for the morphogenesis of the vascular and lymphatic vessels during embryonic development. Heterozygous AM knockout mice survive to adulthood but exhibit elevated blood pressures and an impaired cardiovascular system with increased frequency of heart failure, renal damage, atherosclerosis, and tissue fibrosis. By contrast, mice that overexpress vascular-specific AM suffer from decreased blood pressure and resistance to organ damage and atherosclerosis [8,9,10]. The AM receptor is composed of heterodimers of a GPCR, calcitonin receptor-like receptor (CLR/CALCRL), and a single transmembrane protein, receptor activity modifying protein 2 or 3 (RAMP2 or -3) [11,12,13]. Analysis of vascular endothelial cell-specific *Ramp2* knockout mice revealed that the AM-RAMP2 system is a key determinant of vascular integrity and organ homeostasis [14,15].

Exogenous AM increases renal blood flow, urine volume, and urinary sodium excretion in canine experiments [16]. In addition, AM suppressed some renal damage seen in mice with renal ischemia-reperfusion [10], in a rat model of contrast-induced nephropathy [17], and in a model of renal tubulointerstitial fibrosis due to unilateral ureteral obstruction [18]. In vitro, AM inhibited the cell proliferation of rat glomerular mesangial cells [19]. These results indicate that AM is a potential therapeutic agent for renal damage.

In developing AM as a therapeutic for renal damage, there are concerns that blood AM levels may be elevated by the presence of chronic kidney disease (CKD) or a decrease in renal glomerular filtration rate due to renal damage. In patients with CKD, plasma AM concentrations increase up to 2–3 times those measured in healthy subjects [20]. Continuous intravenous AM infusion in patients with IgA nephropathy increased plasma AM concentrations by approximately two times compared with healthy controls in other reports by the same group [21,22]. The effect of renal dysfunction on blood AM kinetics is not well understood. The aim of this study is, therefore, to investigate the pharmacokinetic changes of AM in two models of rats with renal dysfunction and to further elucidate the mechanism that underlies these changes.

## 2. Materials and Methods

### 2.1. Chemicals

Human AM (hAM, molecular weight 6028.7) was purchased from the Peptide Institute (Osaka, Japan). The other reagents were purchased from FUJIFILM Wako Pure Chemical Corporation (Osaka, Japan).

### 2.2. Animals

Four-month-old male Lewis rats were purchased from Japan SLC (Shizuoka, Japan) and tamoxifen-inducible vascular endothelial cell-specific *Ramp2* knockout mice (E-RAMP2 KO) [23] were the kind gift of Dr. Takayuki Shindo (Shinshu University). Animals were housed under standard laboratory conditions (23 ± 2 °C, 12 h light-dark cycles) and provided with water and chow ad libitum in the Laboratory of Animal Experiments at National Cerebral and Cardiovascular Center (NCVC). All experimental procedures were carried out in accordance with the Regulations for Animal Experimentation and reviewed by the Institutional Animal Care and Use Committee of the NCVC. Based on the national and the institutional regulations and guidelines, all procedures of animal experiments were reviewed by the Committee for Animal Experiments and finally approved by the president of the NCVC. Animals were euthanized at the end of the experiment by the overdose of isoflurane.

### 2.3. Renal Dysfunction Rats with Mercury Chloride Treatment

Renal dysfunction was induced by subcutaneous injection of 3 mg/kg mercury chloride (1 mg/mL saline) to rats (RD-Ag, *n* = 7) on the day before the adrenomedullin pharmacokinetic study [24]. Control rats were treated with saline (normal renal function, NF, *n* = 5). Twenty-four hours after injection of mercury chloride (4 h before hAM administration), blood was collected from the tail vein of rats and serum creatinine (sCr) was measured using DRI-CHEM (FUJIFILM, Tokyo, Japan) to confirm the severity of renal damage. After the hAM administration study, the lungs and kidneys were removed from each group of rats, immersed in RNAlater (Thermo Fisher Scientific, Waltham, MA, USA), and frozen at −30 °C for later analysis.

### 2.4. Renal Dysfunction Model Rats with Bilateral Renal Blood Flow Blockage

Rats were subjected to cannulation at the beginning as described below. After laparotomy through a midline abdominal incision, the bilateral renal artery and vein of the rat were exposed and blocked with vascular clips to develop another model of renal dysfunction (RD-Bl, *n* = 3). Immediately after the vascular blockade, the rats were studied for hAM infusion experiment.

### 2.5. Human Adrenomedullin Infusion Study in Rats

Rats under isoflurane anesthesia were placed on a heated surgical table in the supine position and subjected to cannulation (PE 50, Becton, Dickinson and Company, Franklin Lakes, NJ, USA) of the carotid artery and jugular vein. Human AM was dissolved in 20% mannitol to a concentration of 300 ng/mL and continuously administered at 0.1 µg/kg/min for 60 min via venous cannula using an infusion pump (KDS-100, KD Scientific, Holliston, MA, USA). After hAM administration had begun, 0.6 mL of whole blood was drawn from the arterial cannula at 0, 5, 10, 20, 30, 45, 60, 61.5, 63, 65, 70, 75, 80, and 90 min.

### 2.6. Human Adrenomedullin Intraperitoneal Administration Study in E-RAMP2 KO

E-RAMP2 KO mice (aged 9–10 weeks) were generated by intraperitoneal administration of 75 mg/kg tamoxifen suspended in corn oil (20 mg/mL) once daily for five consecutive days. Mice of the same strain treated only with corn oil were used as a control group. Three to four mice were prepared for each blood collection point. Two weeks after the first dose of tamoxifen, E-RAMP2 KO or control mice were administered 50 μg/kg of hAM intraperitoneally, and 0.6 mL of whole blood was drawn from the inferior vena cava under isoflurane anesthesia before (0 min timepoint) or 10 or 30 min after hAM administration.

### 2.7. Measurement of AM in Plasma

To measure plasma hAM concentrations, whole blood was collected with 2 mg/mL EDTA-2Na and 500 Kallikrein Inhibitory Units (KIU)/mL of aprotinin. Plasma was obtained by centrifugation at 10,000× *g* for 2 min at 4 °C and stored at −30 °C until the assays were performed. Plasma hAM concentration was measured using an automated immunoassay system (AIA-System, Tosoh Corporation, Tokyo, Japan) that was described previously, including intra- and inter-assay variations [25]. This system is a one-step fluorescence sandwich immunoassay for the detection of human AM using two monoclonal antibodies against human AM(12–25) and AM(46–52) [26]. This immunoassay system did not cross-react with either rat or mouse Adm.

### 2.8. Pharmacokinetic Analysis

Pharmacokinetic parameters in rats were calculated from individual plasma AM concentration vs time curves during and after 1 h intravenous infusion of hAM based on standard procedures. The area under the plasma concentration vs time curve (AUC_0–60_) was calculated during the first 60 min using a linear trapezoidal method, and the total systemic clearance (CL_total_) was calculated as the dose administered divided by AUC_0–60_. A plot of plasma concentration vs time after stopping hAM infusion yielded a biexponential curve. The elimination rate constant and half-life (T_1/2β_) in the terminal phase were calculated from individual plasma AM concentration vs time curves with a two-compartment model using the Solver tool in Microsoft Excel to minimize the sum of the squared residual values. Weighting was set to the standard deviation of plasma concentrations measured at each blood sampling point.

### 2.9. mRNA Expression Analysis by Quantitative RT-PCR

All tissues were homogenized in 1 mL TRIzol (Thermo Fisher Scientific) using a TissueLyser II (QIAGEN, Hilden, Germany). Total RNA was purified with an RNeasy Mini Kit (QIAGEN). *Calcrl* and *Ramp2* mRNA levels were determined by quantitative RT-PCR using a LightCycler 480 System II (Roche Applied Science, Indianapolis, IN) and One-Step SYBR PrimeScript TR-PCR kit II (Takara Bio, Shiga, Japan). The following primers were used: *Calcrl,* sense 5′-GTTACACACCAAGCGGAATCCAATC-3′ and antisense 5′-CTTCTCAGAATTGCTTGAACCTCTC-3′; *Ramp2*, sense 5′-CTGGGCTTCCCCAATCCCTTGGCAG-3′and antisense 5′-GGGATGAGGCAGATGGGGGCTATGA-3′; *36B4*, sense 5′-TCATTGTGGGAGCAGACAATGTGGG-3′and antisense 5′-AGGTCCTCCTTGGTGAACACAAAGC-3′. Transcript levels of each gene were normalized against the level of *36B4* mRNA.

### 2.10. Statistical Analysis

All data are presented as means ± SD. Data were analyzed with unpaired t-test for paired data. Non-normally distributed data were analyzed with Mann–Whitney u test (GraphPad, Prism 8, San Diego, CA, USA). We considered *p* values below 0.05 statistically significant. The degree of correlation between continuous variables in rats was calculated using Pearson’s correlation coefficient.

## 3. Results

### 3.1. Plasma Concentration of Exogenous AM Is Elevated in Rats with Mercury Chloride-Induced Renal Dysfunction Due to Decreased Total Systemic Clearance of AM

After subcutaneous administration of mercury chloride to rats to induce renal dysfunction (RD-Ag), the plasma concentration of hAM administered intravenously at 0.1 µg/min/kg for 60 min was compared with that of normal rats (NF). On the day after mercury chloride administration, sCr levels in RD-Ag were 2.21 ± 059 mg/dL, which was significantly higher than 0.26 ± 0.05 mg/dL in NF (*p* < 0.0001). Sixty minutes after the start of exogenous AM administration, plasma AM concentrations reached almost steady state in RD-Ag rats, but still increased in NF rats. Plasma AM concentrations in both groups decreased quickly after the end of AM administration. (Figure 1A, top and middle). Plasma AM concentration at 60 min after initiating treatment (AM C_60_) was significantly higher in RD-Ag (161.82 ± 88.69 pM) than in NF (48.98 ± 13.60 pM) (*p* < 0.05). We measured a statistically significant positive correlation between AM C_60_ and sCr (Figure 1B, *R*^2^ = 0.6149, *p* < 0.01). AUC_0–60_ during continuous hAM infusion was also significantly increased for RD-Ag (40.0 ± 21.1 pg-min/mL) than for NF (8.7 ± 3.1 pg-min/mL) (Table 1). The total systemic clearance (CL_total_) was significantly lower in RD-Ag (191.7 ± 101.7 mL/min/kg) compared with NF (752.3 ± 230.2 mL/min/kg). However, the T_1/2β_ of plasma hAM disappearance after the infusion was halted showed no significant difference between RD-Ag (22.2 ± 9.8 min) and NF (15.0 ± 1.4 min) rats, and no apparent correlation with sCr (Table 1, Figure 2C). These results suggest that AM in RD-Ag rats may be elevated due to a decrease in CL_total_ independent of renal excretion.

To further investigate the effect of renal excretion on plasma AM concentrations, we measured plasma AM in rats with both their renal artery and vein blocked (RD-Bl) under the same AM infusion protocol (Figure 1A, bottom). AM C_60_ (78.12 ± 10.53 pM) as well as AUC_0–60_, T_1/2β_, and CL_total_ in RD-Bl rats were not significantly changed compared to NF (Table 1).

### 3.2. Expression of AM Receptors in Tissues Is Altered in Rats with Mercury Chloride-Induced Renal Dysfunction

To elucidate the mechanism of total systemic clearance reduction in RD-Ag, we measured changes in the transcription of the genes encoding the AM-specific receptors CLR (*Calcrl*) and RAMP2 (*Ramp2*) in the lungs and kidneys of RD-Ag and NF rats. *Calcrl* expression was significantly lower in the kidneys but not the lungs of RD-Ag rats (Figure 2). By contrast, *Ramp2* expression in the kidney and liver did not change significantly between the two groups.

### 3.3. Plasma Concentrations of Exogenous AM Are Augmented in Vascular Endothelium-Specific Ramp2-Deficient Mice

To determine the effect of depleting AM receptors on AM total systemic clearance, we measured plasma AM concentrations after intraperitoneal administration of hAM to E-RAMP2 KO and control mice (Figure 3). In E-RAMP2 KO, plasma AM concentrations were significantly higher at 10 min (E-RAMP2 KO, 3976.73 ± 380.87 pM; Cont, 1067.20 ± 594.92) and 30 min (E-RAMP2 KO, 758.60 ± 233.80 pM; Cont, 117.50 ± 73.69; *p* < 0.05) after administration.

## 4. Discussion

Blood levels of exogenous AM are reportedly elevated in CKD patients [21]. In the study presented here, we investigated the pharmacokinetics of hAM administered intravenously for 60 min in rats with mercury chloride-induced renal dysfunction (RD-Ag) and compared them with normal renal function rats. Plasma AM concentrations in both groups reached a near-steady state at 60 min of continuous AM administration. Both AM C_60_ and AUC_0–60_ increased approximately three times in RD-Ag rats compared with NF rats. Based on our pharmacokinetic analysis results, the increase in plasma AM levels in RD-Ag may be due to decreased CL_total_ of hAM. Furthermore, there was no significant difference in the T_1/2β_ of plasma AM after continuous administration between RD-Ag and NF rats. These results indicated that the CL_total_ of RD-Ag rats was the result of a decrease in the distribution volume of AM.

The blood concentration of molecules removed by the kidneys increases when renal glomerular filtration rate decreases. As is well known, blood NT-proBNP concentrations increase with decreased renal function (progressive kidney disease) [27]. In the model of renal dysfunction used in this study, RD-Ag rats exhibit acute renal dysfunction due to mercury chloride-induced renal tubular injury [28,29]. The sCr of RD-Ag rats was elevated to 10 times that in NF rats with normal renal function, a result indicative of severe renal dysfunction with almost complete loss of glomerular filtration function.

In this study, a significant correlation was found between AM C_60_ and sCr in RD-Ag and NF rats. Conversely, there was no correlation between T_1/2β_ and sCr in these same rats. In a previous report, there was no significant difference in plasma AM concentration between the renal artery and vein in human vascular catheterization [30]. In rats, T_1/2β_ was found to be approximately 1 min [31]. A further report showed that circulating AM was primarily eliminated from the lungs rather than the kidneys [30,32]. These results also suggest that renal clearance has little effect on blood AM levels. In the present study, RD-Bl, in which bilateral renal vessels were clipped and blood flow was blocked, was created as another model of renal dysfunction. Despite the loss of glomerular filtration, there was no difference in either AM C_60_ or AUC_0–60_ between RD-Bl and NF rats. This result supports that renal clearance had no effect on the increase in blood AM levels and suggests that the increase in plasma AM in RD-Ag is caused by a decrease in the distribution volume of AM secondary to renal dysfunction.

At the 0.1 µg/kg/min dose we used for hAM, the plasma AM C_60_ (RD-Ag average of 161.82 pM, NF of 48.98 pM) is lower than its equilibrium dissociation constant for its receptor (*K*_d_ = 0.3–0.6 nM) [33,34]. Therefore, the administered AM appears to bind almost exclusively to its receptor and undergo receptor-mediated clearance. The AM receptor is made up of CLR and RAMP2 or -3 heterodimeric receptors [11,12]. RAMP2 or -3 is a molecular chaperone required for CLR transport to the cell surface as well as for specifying the affinity of CLR for AM [35]. The distribution of ^125^I-labeled AM in rats was abundantly concentrated in the lungs, with a small amount in the renal cortex [36]. Ramp2 mRNA is higher than Ramp3 mRNA in the lungs [37] and kidneys [38] of rats. AM derivative PulmoBind is a molecular SPECT imaging agent distributed in the human lung and is a specific ligand for AM receptor composed of CLR and RAMP2 [39]. In RD-Ag rats, *Calcrl* expression showed a downward trend in the lungs and was significantly reduced in the kidneys. Furthermore, elevated plasma levels of exogenous AM were observed in mice lacking vascular endothelial–specific RAMP2. These results suggest that the decrease in AM receptors associated with renal dysfunction contributes to the decrease in the AM distribution volume. The effects of renal dysfunction on the functioning of several other organs are well known, but its effects on GPCR function and expression are not well understood [40,41,42,43]. Further studies are needed to elucidate the mechanisms of inter-organ communication in kidney disease, including effects on GPCRs.

## 5. Conclusions

Pharmacokinetic studies of exogenous AM in rats with renal dysfunction induced by mercury chloride resulted in elevated plasma AM compared with normal rats, possibly due to decreased total systemic clearance caused by a decrease in the distribution volume of AM. The blood levels of exogenous AM are reportedly higher in patients with chronic kidney disease. From these results, it should be noted that when AM is administered to patients with impaired renal function, plasma concentrations can be expected to be higher than those in healthy subjects.

## Figures and Tables

**Figure 1 biomolecules-12-01281-f001:**
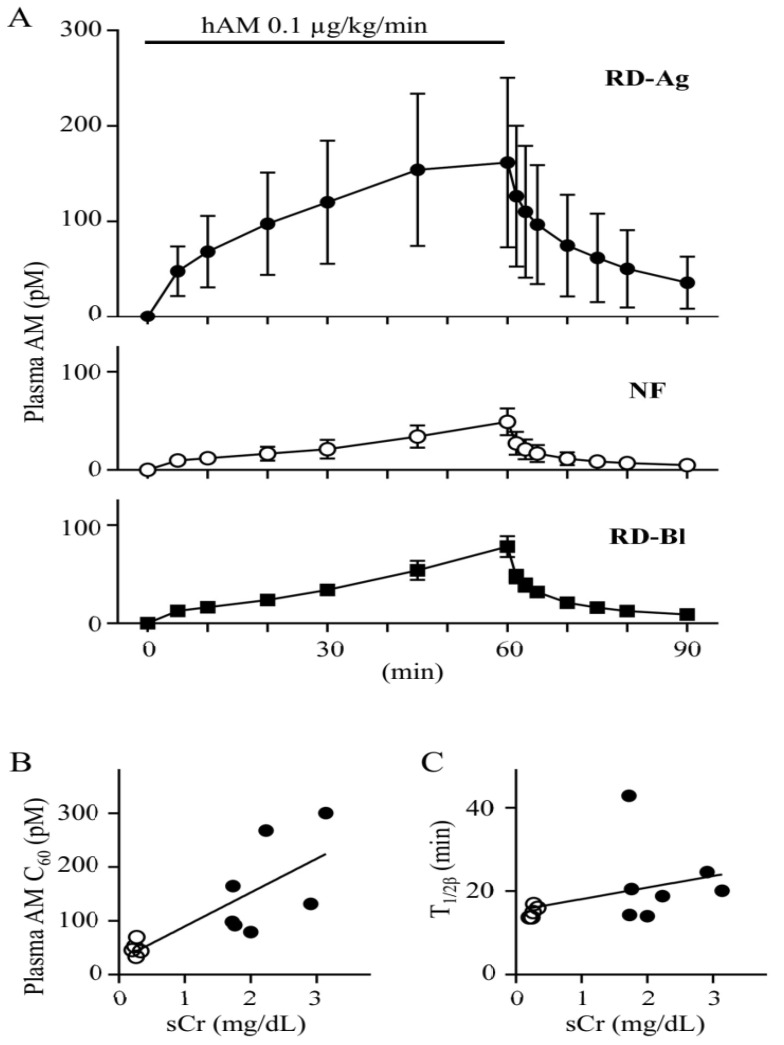
Plasma concentration of human AM in rats with mercury chloride-induced renal dysfunction increases independently of renal excretory function. (**A**) Plasma concentration vs time curve of human adrenomedullin (AM) during 60-min continuous intravenous infusion (0.1 µg/kg/min) in rats with renal dysfunction and normal rats. Closed circles, renal dysfunction rats induced by mercury chloride treatment (RD-Ag, top, *n* = 7); opened circles, normal renal function rats (NF, middle, *n* = 5); closed squares, renal dysfunction rats with bilateral renal blood flow blockage (RD-Bl, bottom, *n* = 3). (**B**) A positive correlation (*R^2^* = 0.6149, *p* < 0.01) between plasma concentration of human adrenomedullin at 60 min for intravenous infusion (AM C_60_) and serum creatinine (sCr) in RD-Ag and NF rats. (**C**) Lack of correlation between plasma elimination half-life (T_1/2β_) and serum creatinine (sCr) in RD-Ag and NF rats. Bars represent means ± SD.

**Figure 2 biomolecules-12-01281-f002:**
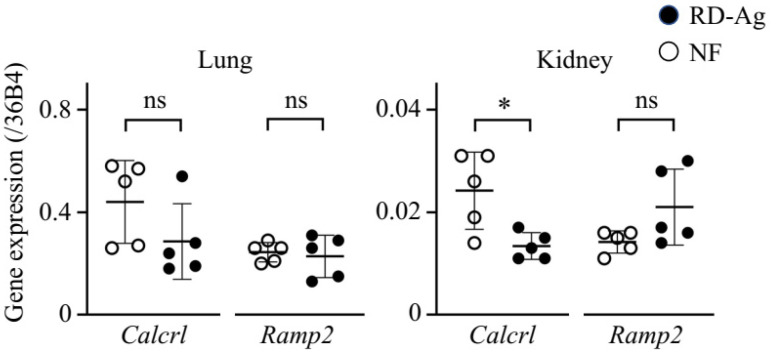
Downregulation of GPCR *Calcrl* gene expression in mercury chloride-treated rats with impaired renal function. Changes in *Calcrl* (calcitonin receptor-like receptor) and *Ramp2* (receptor activity modifying protein 2) mRNA measured by qRT-PCR in the lungs and kidneys of rats with renal dysfunction induced by mercury chloride treatment (RD-Ag, *n* = 5) and normal renal function rats (NF, *n* = 5). Mann-Whitney u test, * *p* < 0.05; ns, not significant. Bars represent means ± SD.

**Figure 3 biomolecules-12-01281-f003:**
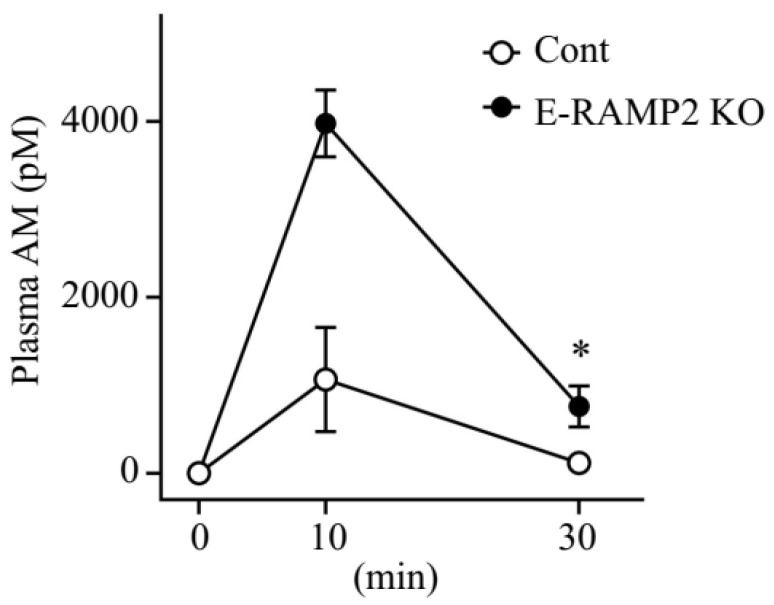
Plasma concentrations of human adrenomedullin administered intraperitoneally are higher in mice deficient in vascular endothelium–specific RAMP2. Changes in plasma concentration after intraperitoneal administration of human adrenomedullin in vascular endothelium–specific Ramp2-deficient mice (E-RAMP2 KO, n = 3–4) and normal controls (Cont, n = 3–4). Mann-Whitney u test, * *p* < 0.05 compared with Cont. Bars represent means ± SD. RAMP2, receptor activity modifying protein 2.

**Table 1 biomolecules-12-01281-t001:** Pharmacokinetic parameters for AM infusion in rats with or without renal dysfunction.

	NF (*n* = 5)	RD-Ag (*n* = 7)	RD-Bl (*n* = 3)
AUC_0–60_ (pg·min/mL)	8.7 ± 3.1	40.0 ± 21.1 **	13.5 ± 1.9
T_1/2β_ (min)	15.0 ± 1.4	22.2 ± 9.8	16.7 ± 1.8
CLtotal (mL/min/kg)	752.3 ± 230.2	191.7 ± 101.7 ***	450.1 ± 68.9

Mean ± SD. ** *p* < 0.01, *** *p* < 0.001 vs NF. NF, normal renal function rats; RD-Ag, rats with renal dysfunction induced by mercury chloride treatment; RD-Bl, rats with renal dysfunction caused by blocking bilateral renal blood flow. AUC_0–60_, area under the plasma concentration vs time curve during the first 60 min; CL_total_, total systemic clearance.

## Data Availability

The data that support the findings of the present study are available from the corresponding author H.H., upon reasonable request.

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
