# Peer review of "Plasma Clearance of Intravenously Infused Adrenomedullin in Rats with Acute Renal Failure"

_biomolecules, 2022, doi:10.3390/biom12091281_

Round 1
Reviewer 1 Report
This article provides insight into how renal function influences the clearance of adrenomedullin in rodents. The paper in general is interesting and well written and provides information that may be useful in the clinic, however there are a number of issues that prevent it from being published in its current format.
Line 18 – The abstract suggests that the plasma levels plateau at this time point, however it does still look like they are increasing linearly in Figure 1.
Line 43 – Emphasis is placed on the cardiovascular phenotype of adrenomedullin knock out, however there is evidence that there is also a lymphatic component to this embryonic lethality. For instance, see Fritz-Six et al., 2008. doi: 10.1172/JCI33302
Line 50 – This line describes adrenomedullin receptors, but completely overlooks CRLR/RAMP3. This is known to be an adrenomedullin receptor (for primary data Hay 2004, https://doi.org/10.1385/JMN:22:1-2:105; for review Hay 2018 doi: 10.1111/bph.14075). Also see the international union of pharmacology:guide to pharmacology database.
Line 68 – sentence should read: two models of rats with renal dysfunction
Line 76 (Section 2.2) – There is no reference to how mice were housed, nor how the RAMP2 KO was generated (e.g. tamoxifen injection? In food?). Was there an end point for euthanisation?
Line 96 – what temperature was the RNA frozen?
Line 101 – Was there a “sham” operation group here? Or is this just compared to control?
Line 124 – please reference the original development of this antibody, as some of the statements in this paragraph require this. This is especially relevant for the statement about the antibody not detecting rat adrenomedullin, as human adrenomedullin and rat adrenomedullin are extremely similar.
Line 167 – There is no reference to rat adrenomedullin. Surely this would be a complicating factor? If human adrenomedullin levels are staying raised, what is happening to rat adrenomedullin?
Line 178 – More explanation as to the rationale would be useful here – I think this method totally removes renal excretion but that is only inferred.
Section 3.2 – A lot is lost by not investigating RAMP3, which is a component of a second adrenomedullin receptor. This either needs to be investigated, or mentioned as a limitation to the study.
Figure 2. Showing the individual data points instead of/in addition to the bars would be useful for understanding how robust this change was. Why is the n number of the RD-Ag rats in this group smaller than the n number of RD-AG rats presented in figure 1.
Section 3.3 – Some rationale would be useful here. How does this relate to the lung clearance hypothesis?
Line 215 – please check whether this is rat or mice. If rats have been compared to mice this is entirely unacceptable.
Figure 3 – Why is there variation of 3-4 n? Which time points are 3 and which are 4? Given that there are multiple comparisons being performed, a two-way ANOVA is more appropriate than multiple t-tests.
Line 252 – Please reference prior knowledge around adrenomedullin being primarily cleared via the lungs rather than renal methods (i.e. Ornan 2002, doi: 10.1016/s0925-4439(01)00108-9.)
Line 262-263 – There is again no reference to the second adrenomedullin receptor (CRLR:RAMP3). This needs to at least be acknowledged, and preferably investigated.
Line 265 – Please reference the fact this is also the case in humans such as with PulmoBind (Harel 2007, https://doi.org/10.1007/s00259-017-3655-y)
Line 282 – How often is adrenomedullin actually administered to renal patients? Is this something that is likely in the future? Please give some context
Author Response
Response to Reviewer 1:
This article provides insight into how renal function influences the clearance of adrenomedullin in rodents. The paper in general is interesting and well written and provides information that may be useful in the clinic, however there are a number of issues that prevent it from being published in its current format.
We appreciate the reviewer’s precise review as well as detailed and insightful comments. We answered and responded to all the comments and points the reviewer described and completely revised the manuscript.
Line 18 – The abstract suggests that the plasma levels plateau at this time point, however it does still look like they are increasing linearly in Figure 1.
Response: We appreciate the reviewer's comment on this point. We have changed “nearly plateaued” to “were still increasing slightly”.
Line 43 – Emphasis is placed on the cardiovascular phenotype of adrenomedullin knock out, however there is evidence that there is also a lymphatic component to this embryonic lethality. For instance, see Fritz-Six et al., 2008. doi: 10.1172/JCI33302.
Response: We appreciate the reviewer's comment on this point. We have added the reference (Ref. 7) and changed the text to “AM is indispensable for the morphogenesis of the vascular and lymphatic vessels during embryonic development.” (line 43).
Line 50 – This line describes adrenomedullin receptors, but completely overlooks CRLR/RAMP3. This is known to be an adrenomedullin receptor (for primary data Hay 2004, https://doi.org/10.1385/JMN:22:1-2:105; for review Hay 2018 doi: 10.1111/bph.14075). Also see the international union of pharmacology:guide to pharmacology database.
Response: In accordance with the reviewer's comments, we have added the reference (Ref. 13) and changed “receptor activity modifying protein (RAMP) to “receptor activity modifying protein 2 or 3 (RAMP2 or -3)” (line 50).
Line 68 – sentence should read: two models of rats with renal dysfunction.
Response: In accordance with the reviewer's comments, we have changed them.
Line 76 (Section 2.2) – There is no reference to how mice were housed, nor how the RAMP2 KO was generated (e.g. tamoxifen injection? In food?). Was there an end point for euthanisation?
Response: In accordance with the reviewer's comments, we have changed “Rats” to “Animals” (line 78). We also added the euthanasia method “Animals were euthanized at the end of the experiment by the overdose of isoflurane.” in line 86.
The method for generating RAMP2 KOs is described in line 111, Section 2.6 and anesthesia method “… the inferior vena cava under isoflurane anesthesia before …” is additionally described in line 117.
Line 96 – what temperature was the RNA frozen?
Response: In accordance with the reviewer's comments, we have added the frozen temperature “at -30°C” in line 96.
Line 101 – Was there a “sham” operation group here? Or is this just compared to control?
Response: We thank the reviewer for this pertinent comment. RD-Bl was just compared to control.
Line 124 – please reference the original development of this antibody, as some of the statements in this paragraph require this. This is especially relevant for the statement about the antibody not detecting rat adrenomedullin, as human adrenomedullin and rat adrenomedullin are extremely similar.
Response: We appreciate the reviewer’s helpful comment. Accordingly, we have added “This system is a one-step fluorescence sandwich immunoassay for the detection of human ADM using two monoclonal antibodies against human ADM(12-25) and ADM(46-52) (Ref. 26).” in line 124.
Line 167 – There is no reference to rat adrenomedullin. Surely this would be a complicating factor? If human adrenomedullin levels are staying raised, what is happening to rat adrenomedullin?
Response: We appreciate the reviewer's comment on this point. Because we were not able to measure rat plasma ADM, the influence of blood ADM on human ADM kinetics in RD-Ag rats is unknown. We agree that additional information on plasma rat ADM in RD-Ag as the reviewer suggested would be valuable. Regrettably, however, because of the lack of rat plasma samples, we are unable to do that additional experimentation.
Line 178 – More explanation as to the rationale would be useful here – I think this method totally removes renal excretion but that is only inferred.
Response: We appreciate the reviewer's concerns on this point. However, we consider that RD-Bl rats may be appropriate model animals for complete suppression of renal excretory function because of their ability to block renal blood flow easily and quickly. Furthermore, the other models of rats with renal dysfunction cannot be denied the possibility that abnormalities of other organs may lead to clearance and degradation of blood adrenomedullin.
we consider our original text correct, because X. Thus, we would like to retain the original text.
since we have already mentioned the pharmacological potential of ghrelin and human clinical studies, we believe that an additional section may not be necessary and would like to leave this point in the original text.
Section 3.2 – A lot is lost by not investigating RAMP3, which is a component of a second adrenomedullin receptor. This either needs to be investigated, or mentioned as a limitation to the study.
Response: We appreciate the reviewer's interest in additional information on RAMP3. However, because the gene expression of Ramp3 is lower than that of Ramp2 in the lung and kidney of rat, we consider that this may not be necessary. We have changed and added the following text (lines 265-266): “The distribution of 125I-labeled AM in rats was abundantly concentrated in the lungs, with a small amount in the renal cortex (Ref. 36). Ramp2 mRNA is higher than Ramp3 mRNA in the lungs (Ref. 37) and kidneys (Ref. 38) of rats. AM derivative PulmoBind is a molecular SPECT imaging agent distributed in the human lung and is a specific ligand for AM receptor composed of CLR and RAMP2 (Ref. 39).”
Figure 2. Showing the individual data points instead of/in addition to the bars would be useful for understanding how robust this change was. Why is the n number of the RD-Ag rats in this group smaller than the n number of RD-AG rats presented in figure 1.
Response: Thank you for offering your valuable comments. We have changed figure 2 according to your comment. In two of the seven RD-Ag rats in figure 1, total RNA could not be obtained from the freeze-preserved tissues, resulting in the analysis of tissue samples from five RD-Ag rats in Figure 2.
Section 3.3 – Some rationale would be useful here. How does this relate to the lung clearance hypothesis?
Response: The results in Section 3.1 and 3.2 suggest that decreased plasma ADM clearance in RD-Ag might be due to a decreased volume of distribution caused by a reduction in adrenomedullin receptors and be not directly related to the lung clearance hypothesis. In Section 3.3, to determine the effect of depleting ADM receptors on ADM total systemic clearance, we measured plasma ADM concentrations after intraperitoneal administration of hADM to E-RAMP2 KO compared with controls.
Line 215 – please check whether this is rat or mice. If rats have been compared to mice this is entirely unacceptable
Response: The reviewer's comment is correct. We have changed “control rats” to “control mice” (line 215).
Figure 3 – Why is there variation of 3-4 n? Which time points are 3 and which are 4? Given that there are multiple comparisons being performed, a two-way ANOVA is more appropriate than multiple t-tests.
Response: Because 0.6 mL of whole blood is required, blood sampling from the same mouse cannot be repeated. Therefore, statistical differences were analyzed by unpaired t-test for paired data. For the sake of being easy to understand, the text has been corrected as follows: Three to four mice were prepared for each blood collection point. Two weeks after the first dose of tamoxifen, E-RAMP2 KO or control mice were administered 50 μg/kg of hADM intraperitoneally, and 0.6 mL of whole blood was drawn from the inferior vena cava under isoflurane anesthesia before (0 min timepoint) or 10 or 30 min after hADM administration (lines 115-117).
Line 252 – Please reference prior knowledge around adrenomedullin being primarily cleared via the lungs rather than renal methods (i.e. Ornan 2002, doi: 10.1016/s0925-4439(01)00108-9.)
Response: We agree with the relevance of this reference and have added “A further report showed that circulating AM was primarily eliminated from the lungs rather than the kidneys (Ref. 30, 32)” to the Discussion and Reference (line 251).
Line 262-263 – There is again no reference to the second adrenomedullin receptor (CRLR:RAMP3). This needs to at least be acknowledged, and preferably investigated.
Response: In accordance with the reviewer's comments, we have changed “RAMP2” to “RAMP2 or -3” (lines 262-263).
Line 265 – Please reference the fact this is also the case in humans such as with PulmoBind (Harel 2007, https://doi.org/10.1007/s00259-017-3655-y)
Response: We appreciate the reviewer’s helpful comment. As recommended, we have added new text “AM derivative PulmoBind is a molecular SPECT imaging agent distributed in the human lung and is a specific ligand for AM receptor composed of CLR and RAMP2 (Ref. 39)” (line 266).
Line 282 – How often is adrenomedullin actually administered to renal patients? Is this something that is likely in the future? Please give some context
Response: AM and its receptors are observed in renal tubular epithelial cells, and AM has been shown to inhibit tubular damage in models of renal tubular injury. In Japan, clinical application of AM for renal disease is under investigation.
Reviewer 2 Report
The manuscript by Hosoda et al. presents a pharmacokinetic study of synthetic human adrenomedullin infused in two rat models of renal dysfunction and in RAMP2 knockout mice. The conclusions are that the higher levels of adrenomedullin found in renal dysfunction may be due to a reduced volume of distribution rather than to glomerular filtration problems.
The methodology and interpretation of the results are adequate but there are a few details that need attention:
1. I am surprised to find the adrenomedullin peptide abbreviated as ADM. Prof. Kitamura, which discovered adrenomedullin, (and many other people following his example) has always abbreviated it as AM. A different question is the gene, which is abbreviated as ADM (Adm in rodents). So, to avoid confusion between the peptide and the gene, I would suggest to use always AM for the peptide.
2. The calcitonin receptor-like receptor is currently abbreviated as CLR (not CRLR).
3. Some English expressions need to be revised throughout the manuscript to convey the proper meaning. For instance “two different rat models of acute renal dysfunction rats” (lines 14-15).
4. Authors claim that adrenomedullin circulates in the blood “at a considerable concentration” (line 40). The concentration presented in reference 5 is just 3.3 femtomoles per milliliter, which is not very “considerable”. Please, rephrase this sentence.
5. In the description of the adrenomedullin receptors (lines 48-50), authors forgot to cite RAMP3 which, although produces a receptor with lower affinity than CLR/RAMP2, is a canonical receptor that may be physiologically relevant. In the same line of thought, I miss RAMP3 expression data on Figure 2. My suggestion is to perform qRT-PCR for RAMP3 in the same samples and add those new data to Figure 2.
6. In the methods section, I´m surprised by the different number of animals used in the various treatments (7 treated with mercury, 5 controls, 3 bilateral blockade). Is there a rational behind this? Did the authors perform a calculation of sample size number?
7. The number of E-RAMP2 KO mice used in the study must be indicated (lines 111-117).
8. In the statistical analysis section (lines 151-154), authors claim they used a t-test. This test can only be used when the datasets follow a normal distribution and are homoscedastic. Since some datasets are composed by very small numbers of subjects (sometimes just 3), there are some doubts about the normality of their distribution. Have the authors performed the proper tests? If so, please include their description.
9. In Results, authors say “both RD-Ag and NF….reached a near stationary state 60 min…” (line 163). This is quite clear for RD-Ag but the graph for NF (and for RD-B1) seems to be still growing fast. Please, rephrase that sentence.
10. Also in the Results section, authors claim “AM C60 was slightly higher in RD-B1…. than in NF rats” (lines 179-180). Apparently, this difference is NOT statistically significant, so the conclusion is that both C60 are undistinguishable. Please, change this sentence.
11. All figures and the text inside them are rather low quality, probably just the printout from the statistics program. It would be appreciated if the authors could provide more professionally drawn figures.
12. Figure legend of Figure 1C begins “Correlation between plasma ….” (line 191). Since this figure presents data that do not correlate, perhaps a better sentence could be “Lack of correlation between plasma….”.
13. In line 215, authors refer to “E-RAMP2 KO and control rats”. I guess they want to say “E-RAMP2 KO and control mice”.
14. The sentence in lines 237-239 is a repetition of the previous one. Please delete one of them.
15. When the authors discuss that renal clearance does not have any effect on AM levels (lines 255-258), they should add that the main organ where AM clearance occurs has been described as the lung. They can use their reference number 27 to support this statement.
16. The authors refer to the KO mice as “mice deleted for vascular … RAMP2” (line 268). A better expression could be “mice deficient in vascular … RAMP2” or “mice lacking vascular … RAMP2”.
Author Response
Response to Reviewer 2:
The manuscript by Hosoda et al. presents a pharmacokinetic study of synthetic human adrenomedullin infused in two rat models of renal dysfunction and in RAMP2 knockout mice. The conclusions are that the higher levels of adrenomedullin found in renal dysfunction may be due to a reduced volume of distribution rather than to glomerular filtration problems.
We wish to express our appreciation for the reviewer’s insightful comments, which have helped me significantly improve the paper. We answered and responded to all the comments and points the reviewer described and completely revised the manuscript.
The methodology and interpretation of the results are adequate but there are a few details that need attention:
- I am surprised to find the adrenomedullin peptide abbreviated as ADM. Prof. Kitamura, which discovered adrenomedullin, (and many other people following his example) has always abbreviated it as AM. A different question is the gene, which is abbreviated as ADM (Adm in rodents). So, to avoid confusion between the peptide and the gene, I would suggest to use always AM for the peptide.
Response: We appreciate the reviewer's comment on this point. As suggested, we have changed ADM to AM throughout the manuscript.
- The calcitonin receptor-like receptor is currently abbreviated as CLR (not CRLR).
Response: In accordance with the reviewer's comments, we have changed CRLR to CLR throughout the manuscript.
- Some English expressions need to be revised throughout the manuscript to convey the proper meaning. For instance “two different rat models of acute renal dysfunction rats” (lines 14-15).
Response: In accordance with the reviewer's comments, we have corrected it to “two models of rats with renal dysfunction” (line 14, 68).
- Authors claim that adrenomedullin circulates in the blood “at a considerable concentration” (line 40). The concentration presented in reference 5 is just 3.3 femtomoles per milliliter, which is not very “considerable”. Please, rephrase this sentence.
Response: The reviewer's comment is correct. We have changed “considerable” to “small” (line 40).
- In the description of the adrenomedullin receptors (lines 48-50), authors forgot to cite RAMP3 which, although produces a receptor with lower affinity than CLR/RAMP2, is a canonical receptor that may be physiologically relevant. In the same line of thought, I miss RAMP3 expression data on Figure 2. My suggestion is to perform qRT-PCR for RAMP3 in the same samples and add those new data to Figure 2.
Response: In accordance with the reviewer's comments, we have changed “receptor activity modifying protein (RAMP) to “receptor activity modifying protein 2 or 3 (RAMP2 or -3)” (line 50).
We appreciate the reviewer's interest in additional information on RAMP3. However, because the gene expression of Ramp3 is lower than that of Ramp2 in the lung and kidney of rat, we consider that this may not be necessary. We have changed and added the following text (lines 265-266): “The distribution of 125I-labeled AM in rats was abundantly concentrated in the lungs, with a small amount in the renal cortex (Ref. 36). Ramp2 mRNA is higher than Ramp3 mRNA in the lungs (Ref. 37) and kidneys (Ref. 38) of rats. AM derivative PulmoBind is a molecular SPECT imaging agent distributed in the human lung and is a specific ligand for AM receptor composed of CLR and RAMP2 (Ref. 39).”
- In the methods section, I´m surprised by the different number of animals used in the various treatments (7 treated with mercury, 5 controls, 3 bilateral blockade). Is there a rational behind this? Did the authors perform a calculation of sample size number?
Response: We agree with the reviewer's comments. However, because the adrenomedullin kinetics obtained in the three RB-Bl rats were not varied and did not differ from those in the NF rats, no additional RB-Bl rat experiments were performed.
- The number of E-RAMP2 KO mice used in the study must be indicated (lines 111-117).
Response: We appreciate the reviewer’s helpful comment. We have added new text “Three to four mice were prepared for each blood collection point.” in line 115.
- In the statistical analysis section (lines 151-154), authors claim they used a t-test. This test can only be used when the datasets follow a normal distribution and are homoscedastic. Since some datasets are composed by very small numbers of subjects (sometimes just 3), there are some doubts about the normality of their distribution. Have the authors performed the proper tests? If so, please include their description.
Response: In accordance with the reviewer's comment, we have changed text to “Data were analyzed with unpaired t-test for paired data. Non-normally distributed data were analyzed with Mann-Whitney u test (GraphPad, Prism 8, San Diego, CA).” (lines 151-152).
We have change “Unpaired t-test” to “Mann-Whitney u test” in line 210.
And we have change “Unpaired t-test, **p < 0.01” to “Mann-Whitney u test, *p < 0.05” in lines 223-224.
- In Results, authors say “both RD-Ag and NF….reached a near stationary state 60 min…” (line 163). This is quite clear for RD-Ag but the graph for NF (and for RD-B1) seems to be still growing fast. Please, rephrase that sentence.
Response: In accordance with the reviewer's comments, we have corrected them to “Sixty minutes after the start of exogenous AM administration, plasma AM concentrations reached almost steady state in RD-Ag rats, but still increased in NF rats. Plasma AM concentrations in both groups decreased quickly after the end of AM administration.” (lines 162-164).
- Also in the Results section, authors claim “AM C60 was slightly higher in RD-B1…. than in NF rats” (lines 179-180). Apparently, this difference is NOT statistically significant, so the conclusion is that both C60 are undistinguishable. Please, change this sentence.
Response: In accordance with the reviewer's comments, we have change them to “AM C60 (78.12 ± 10.53 pM) as well as AUC0–60, T1/2β, and CLtotal in RD-Bl rats were not significantly changed compared to NF (Table 1)” (lines 179-181).
- All figures and the text inside them are rather low quality, probably just the printout from the statistics program. It would be appreciated if the authors could provide more professionally drawn figures.
Response: Thank you for offering your valuable comments. We have changed all figures according to your comment.
- Figure legend of Figure 1C begins “Correlation between plasma ….” (line 191). Since this figure presents data that do not correlate, perhaps a better sentence could be “Lack of correlation between plasma….”.
Response: We appreciate the reviewer’s helpful comment. As recommended, we have changed to “Lack of correlation between plasma…” (line 191).
- In line 215, authors refer to “E-RAMP2 KO and control rats”. I guess they want to say “E-RAMP2 KO and control mice”.
Response: The reviewer's comment is correct. We have changed “control rats” to “control mice” (line 215).
- The sentence in lines 237-239 is a repetition of the previous one. Please delete one of them.
Response: In accordance with the reviewer's comments, we have removed the later text (lines 237-239).
- When the authors discuss that renal clearance does not have any effect on AM levels (lines 255-258), they should add that the main organ where AM clearance occurs has been described as the lung. They can use their reference number 27 to support this statement.
Response: We agree with the relevance of this reference and have added “A further report showed that circulating AM was primarily eliminated from the lungs rather than the kidneys (Ref. 30, 32)” to the Discussion and Reference (line 251).
- The authors refer to the KO mice as “mice deleted for vascular … RAMP2” (line 268). A better expression could be “mice deficient in vascular … RAMP2” or “mice lacking vascular … RAMP2”.
Response: We appreciate the reviewer’s helpful comment. We have corrected to “mice lacking vascular … RAMP2” (line 268).
Round 2
Reviewer 1 Report
Many thanks to the authors for these corrections and answers. I am now happy for this to be published in this format.
Author Response
Thank you very much for providing important comments. We are thankful for the time and energy you expended.
Reviewer 2 Report
I appreciate the modifications made to the manuscript that clearly improved its quality.
I still think that looking for RAMP3 expression in the samples was not a Herculean task, but the authors chose not to do it. Probably the contribution of RAMP3 is not major in these models, but showing the data is always better than guessing.
The quality of the Figures has clearly improved. Nevertheless, Figures 1 and 3 still abbreviate adrenomedullin as ADM, which is confusing when the abbreviation in the text was changed to AM. Please modify the figures accordingly.
Author Response
Response to Reviewer 2:
I appreciate the modifications made to the manuscript that clearly improved its quality.
I still think that looking for RAMP3 expression in the samples was not a Herculean task, but the authors chose not to do it. Probably the contribution of RAMP3 is not major in these models, but showing the data is always better than guessing.
Thank you very much for providing important insights. We are grateful for the time and energy you expended on our behalf.
The quality of the Figures has clearly improved. Nevertheless, Figures 1 and 3 still abbreviate adrenomedullin as ADM, which is confusing when the abbreviation in the text was changed to AM. Please modify the figures accordingly.
In accordance with the reviewer's comments, we have changed ADM to AM in Figure1 and 3.